# Weak-to-strong Generalization via Formative Learning from Student Demonstrations & Teacher Evaluation

## Abstract

As Large Language Models (LLMs) exceed human capabilities, providing reliable human feedback for evaluating and aligning them, via standard frameworks such as Reinforcement Learning from Human Feedback, becomes challenging. This raises a fundamental question: *how can we leverage weaker (teacher) supervision to elicit the full capabilities of a stronger (student) model?* This emerging paradigm, known as Weak-to-Strong (W2S) generalization, however, also introduces a key challenge as the strong student may "overfit" to the weak teacher's mistakes, resulting in a notable performance degradation compared to learning with ground-truth data. We show that this overfitting problem occurs because learning with weak supervision implicitly regularizes the strong student's policy toward the weak reference policy. Building on this insight, we propose a novel learning approach, called Weak Teacher **Ev**aluation of Strong Student **De**monstrations or EVE, to instead regularize the strong student toward its reference policy. EVE's regularization intuitively elicits the strong student's knowledge through its own task demonstrations while relying on the weaker teacher to evaluate these demonstrations – an instance of formative learning. Extensive empirical evaluations demonstrate that EVE significantly outperforms existing W2S learning approaches and exhibits significantly better robustness under unreliable feedback compared to contrastive learning methods such as Direct Preference Optimization.

## 1 Introduction

Reinforcement Learning from Human Feedback (RLHF) [23, 5] has been a canonical framework for steering language models (LMs) to align with human values based on human demonstrations. This framework has demonstrated impressive performance across a wide range of tasks, from conversation to coding, where humans "can" provide reliable supervision. In the future, as these AI models reach or exceed human capabilities, they will be capable of solving complex tasks that are difficult for humans to supervise. For example, when these AI models acquire the ability to generate a code project with millions of lines of code or summarize an entire book with thousands of pages, humans are unlikely to provide reliable feedback to align these superhuman AI models effectively.

*How can we align these superhuman AI models given the likely unreliable human supervision?* Burns et al. [4] study this question by using a smaller LLM to represent unreliable human supervision on binary classification tasks. Effectively, this "weaker" teacher is prone to make mistakes when supervising a "stronger" student model. They observed a phenomenon called *weak-to-strong (W2S) generalization* – a stronger model finetuned with labels generated by a weaker model could outperform this weaker teacher without even seeing the ground truth labels. Despite the promising results, a key challenge in learning from weak supervision is the risk of overfitting [4], where the strong student inevitably learns to imitate the errors of the weak teacher. Burns et al. [4] study early-stopping as an

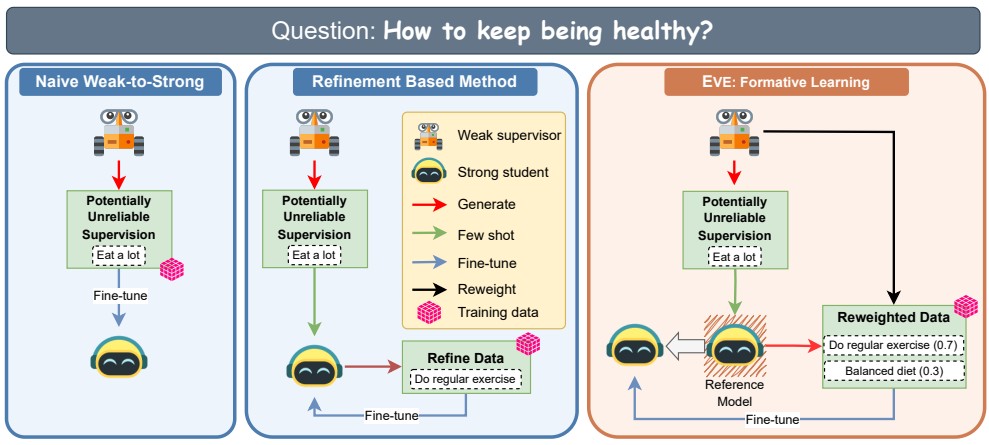

Figure 1: **EVE and existing W2S generalization methods**. Naive learning overfits the weak reference model, potentially imitating its mistakes (e.g., *"Eat a lot"*). Refinement learning "refines" the weak supervision (i.e., *"Do regular exercise"*). In contrast, EVE leverages the weak teacher as a reward function while eliciting the student's reference model salient knowledge

implicit regularization to prevent overfitting, but notes that early-stopping does not constitute a valid method as it unrealistically requires ground-truth labels.

This paper first provides a crucial theoretical insight into the overfitting problem in W2S generalization. Specifically, by representing the weak teacher as an Energy-Based Model (EBM), we reveal that learning from weak supervision involves maximizing the reward while simultaneously regularizing the strong student's policy toward the weak reference model. This process leads to a drawback: the strong student not only inherits the informative supervision but also amplifies the errors of the weak teacher, ultimately degrading the student's overall performance on the desired tasks [14].

Building upon this insight, we propose a novel learning method, called Weak Teacher **Ev**aluation of Strong Student **De**monstrations (EVE), to enable the strong student to elicit its own (prior) knowledge of the task while relying on the weak teacher to evaluate, or score, such demonstrations – an instance of formative learning, effectively utilizing both the knowledge of the weak teacher and the student's reference model. As depicted in Fig. 1, EVE utilizes the weak teacher's demonstrations to prompt the strong student, allowing it to generate its own training data reflecting its understanding of the tasks. The generated samples are then adjusted by the logarithmic ratio of the weak teacher's policy pre- and post-alignment, which serves as a reward signal to guide the strong student's learning.

In summary, (1) we provide a theoretical characterization of overfitting in W2S learning; then (2) we introduce EVE, an approach that enables learning from strong student demonstrations, where the weak teacher acts as a reward function to evaluate the strong student's outputs; finally, (3) we show that EVE significantly outperforms naive W2S learning by overcoming the overfitting issue, demonstrating the effectiveness of utilizing the strong student's critical thinking ability under the weak teacher's reward evaluation; surprisingly, when learning from a weak and unreliable reward signal, EVE – an off-policy method – achieves significantly better performance and robustness to contrastive learning methods such as DPO [30].

## 2 Related Work

### 2.1 Weak-to-strong Generalization

Burns et al. [4] introduce a synthetic setup to study whether a stronger model can generalize well with weaker supervision, compared to training with high-quality or ground-truth data. Prior efforts investigate W2S phenomena only in binary classification setups, leaving other practical alignment-relevant tasks (e.g., open-ended text generation whose output has no fixed length and requires sharing vocabulary size between the strong student and weak teacher) largely under-explored [43, 6, 1]. Another line of work [34, 44, 45] leverages the pre-trained knowledge of the strong student to refine

labels curated from the weak teacher, thereby improving the supervision quality. Ye et al. [44] study W2S generalization on text-generation tasks, where they simulate *unreliable demonstrations* and *unreliable comparison feedback* during the alignment phase.

Different from the prior work, this paper extends W2S generalization beyond classification. We elicit the latent knowledge of the strong student about the intended tasks, which is then evaluated by the weak teacher's reward model. Additionally, by interpreting learning from weak supervision as reward maximization, our approach generalizes refinement-based methods [44, 41].

'

## 2.2 Reinforcement Learning from Human Feedback

RLHF aims to align LMs with human preferences and values [5, 3], and has demonstrated impressive performance on established benchmarks [22, 15, 39, 40, 38]. However, the RLHF pipeline incurs significant computational costs and requires a large amount of high-quality human preference labels.

Recent advancements, such as Direct Alignment Algorithms (DAAs) [30, 37], bypass the need for an explicit reward model and directly train the LMs on the human preference data. Reinforcement Learning with AI Feedback [25] uses a well-trained language model (e.g., GPT-4 or Claude-3.5 Sonnet) to provide preference feedback as a substitute for human supervision. More recently, Ye et al. [44] study whether standard RLHF remains effective under unreliable feedback.

We demonstrate that **contrastive-learning** approaches [31, 2] heavily suffer from the reward over-optimization issue [31, 10]. In contrast, EVE – also an offline supervised approach – is significantly more robust to unreliable feedback and achieves a better reward-KL tradeoff than DAAs. This finding is significant as it contradicts observations in prior work [36], which shows that DAAs with negative gradient perform significantly better than offline supervised methods in conventional alignment scenarios with human feedback.

## 2.3 Reward Maximization with KL Regularization as Distributional Matching

Prior works show that reward maximization with KL regularization in standard RLHF can be viewed as minimizing the reverse KL between the LM policy $\pi_\theta$ and the target distribution that represents the aligned language model [17, 16, 12]. Other studies also explored the use of forward KL, which corresponds to setting the reward maximization as supervised learning [21, 28]. Similarly, our paper shows that imitating a weak teacher can be viewed as reward maximization, where the reward is defined as the log probability of the weak teacher, with a KL regularization toward the weak reference model, causing the over-optimization problem.

# 3 Preliminaries

## 3.1 LLM Alignment with Human Preferences

LLM alignment can be viewed as reward-maximization with KL-constrained:

$$\max_{\pi_\theta} \mathbb{E}_{x \sim \mathcal{D}, y \sim \pi_\theta(\cdot|x)} \left[ r(x, y) \right] - \beta \text{KL}(\pi_\theta || \pi_{\text{ref}}) \tag{1}$$

where $y$ is a sampled response from $\pi_\theta$, $\beta$ controls the trade-off between maximizing the reward and deviation from the reference model $\pi_{\text{ref}}$, and $r$ is the reward function that captures human preferences.

## 3.2 Offline Fine-Tuning Methods for Reward Maximization

Directly optimizing the objective in Eq. (1) requires repeated sampling, which is computationally expensive. Alternatively, equivalent offline methods fall into 2 main categories:

**Contrastive Learning Methods.** Approaches, such as DPO [30] and IPO [2], directly update the LM policy $\pi_\theta$ on human preference data. These methods represent the reward implicitly via the LM $\pi_\theta$ and the reference model $\pi_{\text{ref}}$ as:

$$r_\theta(x, y) = \beta \log \frac{\pi_\theta(y|x)}{\pi_{\text{ref}}(y|x)} + \beta \log Z(x) \tag{2}$$

where $Z(x) = \sum_y \pi_{\text{ref}}(y|x) \exp\left(\frac{1}{\beta} r_\theta(x, y)\right)$ is the normalization factor. Using this representation, a general objective can be derived to train the policy on human preference data, as follows:

$$\mathcal{L}(\pi_\theta, \pi_{\text{ref}}) = -\mathbb{E}_{(x,y_w,y_l)\sim\mathcal{D}} \left[ f\left( \beta \log \frac{\pi_\theta(y_w|x)}{\pi_{\text{ref}}(y_w|x)} - \beta \log \frac{\pi_\theta(y_l|x)}{\pi_{\text{ref}}(y_l|x)} \right) \right] \tag{3}$$

where $f$ is a convex loss function. The gradient of contrastive learning approaches, therefore, consists of a **positive gradient** term that increases the likelihood of the preferred response $y_w$ and a **negative gradient** term that pushes down the likelihood of the non-preferred response $y_l$.

**Offline Supervised Methods.** This alternative class of methods, including RAFT [8] and RWR [28], minimizes a weighted maximum likelihood objective. Formally, these methods first sample $K$ completions per prompt $x$ from the reference model $\pi_{\text{ref}}$, i.e., $y_{1,\cdots,K} \sim \pi_{\text{ref}}(\cdot|x^{(i)})$. These responses are then weighted by a non-negative weighting function $F(x, y_k|y_{1,\cdots,K})$ conditioned on the other sampled responses and maximize:

$$\max_{\pi_\theta} \mathbb{E}_{(x,y_1,\cdots K)\sim\mathcal{D}, \pi_{\text{ref}}(\cdot|x)} \left[ \sum_{k=1}^{K} \log \pi_\theta(y_k|x) \cdot F(x, y_k|y_{1,\cdots,K}) \right]$$

Intuitively, since $F(x, y|y_{1,\cdots K})$ is always non-negative, these methods always increase the likelihood of responses generated from $\pi_{\text{ref}}$. Responses that are more preferred will be assigned higher weights, there is no **negative gradient** effect to push down the likelihood of suboptimal responses.

### 3.3 Weak-to-Strong Evaluation Pipeline

We review the W2S evaluation pipeline in [4], which consists of three stages, as follows:

**(1) Weak Teacher Creation:** The weak teacher is created by fine-tuning a small pre-trained model to align with human preferences. We utilize SFT+DPO, a standard preference learning pipeline, to ensure the weak model acquires knowledge about alignment tasks. The resulting model is denoted as $\pi^{\text{weak}}$.

**(2) Strong Student Learning with Weak Supervision:** The weak model is then used to generate weak supervision data $\mathcal{D}_{\text{weak}} = \{x^{(i)}, y^{(i)}\}$ where $x^{(i)}$ and $y^{(i)}$ are the prompt and the generated response from $\pi^{\text{weak}}$, respectively. The strong model $\pi_\theta$ is then fine-tuned using the weak supervision data with the SFT objective.

**(3) Strong Student Learning with Ground-truth Supervision:** Another strong model $\pi^{\text{strong}}$ is fine-tuned with the Ground-truth human labels to establish the upper-bound performance. To ensure that this aligned model fully acquires the target task's capabilities, it goes through an additional, preference learning phase (e.g., DPO).

The W2S generalization performance of $\pi_\theta$ can be measured by Performance Gap Recovered (**PGR**):

$$\text{PGR} = \frac{\mathcal{P}_{\text{weak-to-strong}} - \mathcal{P}_{\text{weak}}}{\mathcal{P}_{\text{strong}} - \mathcal{P}_{\text{weak}}}$$

where $\mathcal{P}_{\text{weak-to-strong}}$, $\mathcal{P}_{\text{weak}}$, and $\mathcal{P}_{\text{strong}}$ are the task performance of $\pi_\theta$, $\pi^{\text{weak}}$, and $\pi^{\text{strong}}$, respectively.

## 4 Formative Learning with EVE

### 4.1 Learning from Weak Supervision Implicitly Aligns with Weak Reference Model

This section connects W2S learning to reward maximization and builds the theory behind the model's behavior, i.e., its generalization characteristics.

We begin by representing the weak teacher in the form of energy-based models [30, 18, 13]:

$$\pi^{\text{weak}}(y|x) = \frac{1}{Z(x)} \pi_{\text{ref}}^{\text{weak}}(y|x) \exp\left(r^{\text{weak}}(x, y)/\beta\right)$$

where $\pi_{\text{ref}}^{\text{weak}}$ is the SFT version of $\pi^{\text{weak}}$.

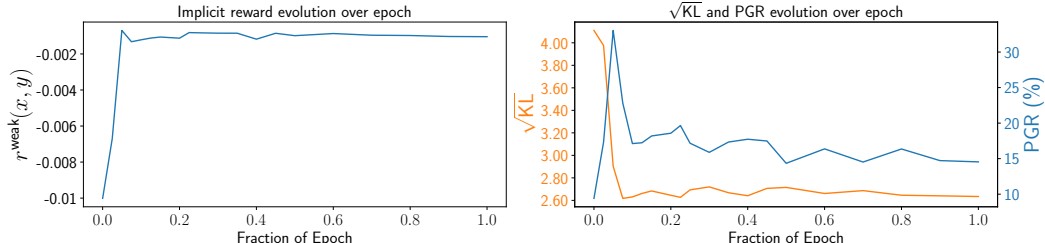

Figure 2: **Learning from weak supervision** as reward maximization. **Left:** the strong model $\pi_\theta$ learns to maximize the implicit reward $r^{\text{weak}}(x, y) = \beta \log \pi_{\text{align}}^{\text{weak}}(y|x)$ - $\beta \log \pi_{\text{ref}}^{\text{weak}}(y|x)$. **Right:** the strong model also learns to imitate the weak reference model $\pi_{\text{ref}}^{\text{weak}}$'s mistakes, leading to performance degradation (in PGR).

**Proposition 4.1.** *W2s generalization with a weak teacher $\pi^{weak}(y|x)$ and a strong student $\pi_\theta$ (the training model) can be cast as the following optimization problem:*

$$\min_{\pi_\theta} KL\left(\pi^{weak}||\pi_\theta\right) \tag{4}$$

$$s.t \ \pi^{weak} = \arg\min_{\pi} KL\left(\pi||\pi^{EBM}\right)$$

*where $\pi^{EBM}(y|x) \propto \pi_{ref}^{weak}(y|x) \exp\left(r(x, y)/\beta\right)$.*

The proof is straightforward and deferred to the Appendix D.2. This shows that imitating the weak teacher can be seen as finding an EBM policy $\pi^{\text{EBM}}$, which is the optimal solution in the lower-level objective. This leads to the following theorem.

**Theorem 4.2.** *The optimal solution to W2S generalization is equivalent to the optimal solution in the following objective:*

$$\max_{\pi_\theta} \mathbb{E}_{x \sim \mathcal{D}, y \sim \pi_\theta(\cdot|x)} \left[r^{weak}(x, y)\right] - \lambda KL(\pi_\theta||\pi_{ref}^{weak}) \tag{5}$$

***Proof Sketch.*** Notice that the objective for training the strong student, and the reverse KL share the same optimal solution $\pi_\theta$. In addition, it can be shown that minimizing the reverse KL between the strong student and the weak teacher,

$$\min_{\pi_\theta} KL\left(\pi_\theta||\pi^{\text{weak}}\right), \tag{6}$$

is equivalent to maximizing the KL-constrained reward objective in Eq. (5). $\qquad\square$

Theorem 4.2 provides a key insight: imitating the weak teacher maximize an implicit reward, $r^{\text{weak}}(x, y) = \beta \log \pi^{\text{weak}}(y|x) - \beta \log \pi_{\text{ref}}^{\text{weak}}(y|x)$, while regularizing (with KL objective) the strong student toward the weak reference model $\pi_{\text{ref}}^{\text{weak}}$. Consequently, instead of aiming to elicit knowledge of the strong student, existing W2S learning remains confined to the knowledge of the weak model, which may adversely impact the strong student's performance.

### 4.2 Suboptimal Weak-to-Strong Generalization toward Weak Reference Model

We empirically confirm the theoretical insight in the previous section. Specifically, we analyze the W2S training progression on $\mathcal{D}_{\text{weak}}$: at each checkpoint, we generate responses using the corresponding intermediate model with the same set of prompts, from which we calculate the implicit reward $r^{\text{weak}}(x, y) = \beta \log \pi^{\text{weak}}(y|x) - \beta \log \pi_{\text{ref}}^{\text{weak}}(y|x)$, the divergence $\text{KL}(\pi_\theta||\pi_{\text{ref}}^{\text{weak}})$, and the PGR.

Fig. 2 shows that while the strong model learns to maximize the implicit reward (Left), the learned policy is also regularized towards the weak reference model $\pi_{\text{ref}}^{\text{weak}}$, indicated by the consistently low KL divergence $\text{KL}(\pi_\theta||\pi_{\text{ref}}^{\text{weak}})$ shortly after the training progresses (Right). Moreover, we also observe that the PGR, as measured by the golden reward function, decreases significantly (Right). This suggests that imitating the weak reference model $\pi_{\text{ref}}^{\text{weak}}$ (and potentially inheriting its mistakes) negatively impacts the performance of the strong student.

### 4.3 EVE: Eliciting Strong Student Knowledge

Motivated by the connection between imitating the weak teacher and reward maximization, we "generalize" the KL-constrained reward maximization learning of the strong student $\pi$:

$$\max_{\pi_\theta} \mathbb{E}_{x\sim\mathcal{D}, y\sim\pi_\theta(\cdot|x)} \left[ r^{\text{weak}}(x,y) \right] - \lambda\text{KL}(\pi_\theta||\hat{\pi}) \tag{7}$$

where $\lambda$ controls the trade-off between maximizing the reward and deviation from a regularization policy $\hat{\pi}(y|x)$. Next, we propose one specific choice of the regularization policy $\hat{\pi}$ that can facilitate the elicitation of the strong student's knowledge, thereby enhancing W2S generalization.

**The choice of regularization policy $\hat{\pi}$.** Burns et al. [4] interpret W2S generalization in terms of saliency: some tasks are already salient to the strong student; in this view, the role of the weak teacher is to elicit the student's latent knowledge rather than enforcing naive imitation of the weak teacher's own demonstrations. Inspired by this interpretation, we propose to regularize the learning policy toward the strong student pre-trained model, i.e., $\hat{\pi}(y|x) = \pi_{\text{ref}}^{\text{strong}}(y|x)$. This design choice serves an important goal: to encourage the learned policy $\pi_\theta$ to remain close to the initial strong reference model $\pi_{\text{ref}}^{\text{strong}}$, thereby facilitating the elicitation of the student's prior knowledge while simultaneously incorporating assessment from the weak teacher. Similar to [4], to elicit the strong student's knowledge of the task, we first create the weak teacher's demonstrations, which are then used in few-shot prompting the strong reference model $\pi_{\text{ref}}^{\text{strong}}$ to generate task-relevant outputs, as $\pi_{\text{ref}}^{\text{strong}}$ is not trained to follow instructions. We provide detailed examples in Appendix B.6.

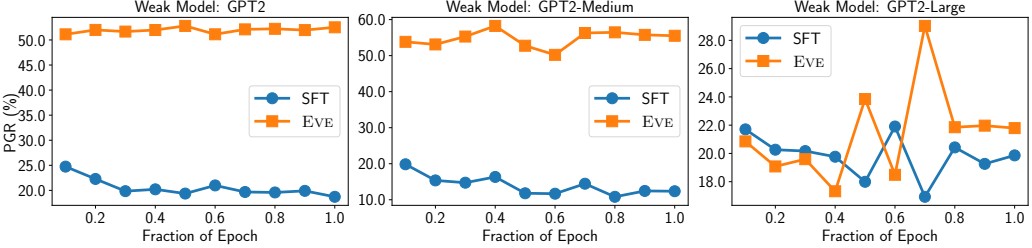

Figure 3: Evolution of PGR (%). We observe clear signs of overfitting to the weak teacher's errors well before finishing a single epoch. Notably, when there is a large gap between the strong student and the weak teacher, the student reaches its best performance within the first 10% of the epoch. EVE has little to no PGR degradation and significantly outperforms naive W2S learning (SFT).

**Optimization.** Directly optimizing the objective in Eq. (7) can incur significant computational costs, as it requires repeated sampling from the strong student $\pi_\theta$ inside the training loop [30]. Following prior work [30, 28, 27], it is straightforward to show that the optimal policy to this KL-constrained objective takes the form:

$$\pi_r(y|x) = \frac{1}{Z(x)} \exp\left(r(x,y)/\lambda\right) \pi_{\text{ref}}^{\text{strong}}(y|x)$$

where $Z(x) = \sum_y \pi_{\text{ref}}^{\text{strong}}(y|x) \exp\left(r(x,y)/\lambda\right)$ is the normalization constant. We can also leverage the duality between the reward function and the weak teacher $\pi^{\text{weak}}$ [30]. Given the optimal policy $\pi_r$, we can then formulate a supervised learning objective for the parametrized strong student $\pi_\theta$ to match with this optimal policy, resulting in the following objective:

$$\max_{\pi_\theta} \mathcal{J}(\pi_\theta) = \max_{\pi_\theta} \mathbb{E}_{x\sim\mathcal{D}, y\sim\pi_{\text{ref}}^{\text{strong}}(\cdot|x)} \left[ \frac{\left(\pi^{\text{weak}}(y|x)/\pi_{\text{ref}}^{\text{weak}}(y|x)\right)^{\beta/\lambda}}{Z(x)} \cdot \log \pi_\theta(y|x) \right]$$

where the $\beta/\lambda$ ratio controls the impact of the weak-supervision reward signal during the strong student's updates. A high $\beta/\lambda$ ratio leads to a more uniform update, where all samples are assigned similar weights; i.e., there will be no weak supervision in learning. Conversely, a low $\beta/\lambda$ ratio results in a more focused policy update that prioritizes samples with high weak-supervision reward signals. This objective avoids sampling directly from $\pi_\theta$ on every update as $\pi_\theta$ changes during

training; instead, we can sample the responses from the fixed $\pi_{\text{ref}}^{\text{strong}}$ once at the beginning of the optimization, which is significantly more efficient.

We also estimate the intractable normalization factor $Z(x)$ using *Self-Normalizing Importance Sampling* [24]. Formally, given $K > 1$ i.i.d. completions $y^1, \cdots, y^N \sim \pi_{\text{ref}}^{\text{strong}}(\cdot|x)$ drawn from strong reference model, we can define an empirical distribution by normalizing the log-ratio $f(x, y) = \frac{\beta}{\lambda}\left(\log \pi^{\text{weak}}(y|x) - \log \pi_{\text{ref}}^{\text{weak}}(y|x)\right)$ over $K$ samples:

$$F(x, y^i|y^{1, \cdots, K}) = \frac{K \cdot \exp\left(f(x, y^i)\right)}{\sum_{k=1}^{K} \exp\left(f(x, y^k)\right)} \tag{8}$$

where the normalization is estimated by $Z(x) \approx \frac{1}{K} \sum_{k=1}^{K} \exp\left(f(x, y^k)\right)$. In summary, the final estimate is:

$$\mathcal{J}(\pi_\theta) = \mathbb{E}_{x \sim \mathcal{D}, y^{1, \cdots, K} \sim \pi_{\text{ref}}^{\text{strong}}(\cdot|x)}\left[\log \pi_\theta(y^i|x) \cdot F(x, y^i|y^{1, \cdots, K})\right]$$

We refer to this W2S learning approach as EVE. EVE can be seen as an offline supervised method, where the weighting function is the exponential of the implicit reward defined in Eq. (2).

## 5 Experiments

In this section, we empirically evaluate EVE's W2S generalization performance on two tasks: **controlled-summarization** and **instruction following**.

### 5.1 Controlled-Summarization

**Setup.** We choose the representative Reddit TL;DR summarization [35] dataset and follow the synthetic setup from [10, 47, 30], where we train a *golden* reward model $r_{\text{gold}}(x, y)$ to label synthetic preference data $\mathcal{D}_{\text{golden}}$ for fine-tune weak-aligned model and evaluation. We use GPT2-series [29] (GPT2-Base/Medium/Large) as weak teachers and a more advanced LLama-3.2-3B model [19, 20] as the strong student. The weak model $\pi^{\text{weak}}$ is the aligned model with DPO [30] from $\mathcal{D}_{\text{golden}}$.

**Baselines.** In additional to EVE, we evaluate several existing W2S approaches, including **SFT** – which naively fine-tunes the strong student on weak supervision data $\mathcal{D}_{\text{weak}}$ – and (2) **Refinement** [34, 41] – which prompts the strong student to refine the responses generated by the weak teacher and fine-tunes the strong student with the refined responses.

**Results.** Fig. 4 shows the PGR results. EVE consistently outperforms the other baselines across all weak teachers. Notably, under the supervision of GPT-2 (the weakest model), EVE achieves a nearly 25% performance boost over SFT. Moreover, SFT achieves the peak performance early in training (around 10% of the epoch), but its performance steadily declines thereafter.
In contrast, **EVE demonstrates minimal to no degradation in PGR over the course of the training process**. As discussed in Section 4, this can be attributed to the ability of EVE to more effectively balance learning from the weak teacher and the salient knowledge of the strong reference model.

**Impact of $\beta/\lambda$ ratio.** We investigate the impact of $\beta/\lambda$ on W2S performance. Fig. 5 illustrates the impact of $\beta/\lambda$ on PGR across different weak teachers. Setting $\beta/\lambda$ around 1.0 achieves optimal or near-optimal performance. Consequently, we default $\beta/\lambda = 1.0$ in all experiments, **eliminating the need for hyperparameter tuning that requires ground-truth labels**. Without the

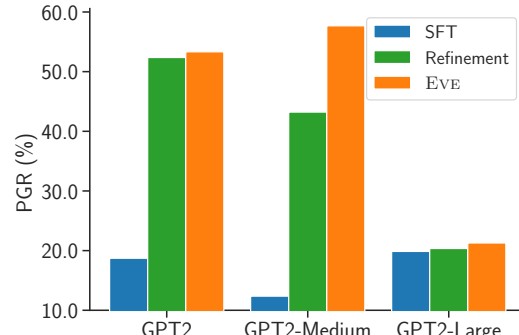

Figure 4: PGR (%) of SFT, Refinement and EVE.

weak supervision (i.e., $\beta/\lambda = \infty$), the performance significantly decreases; this confirms the benefit of learning from the weak teacher's reward signals. Conversely, setting $\beta/\lambda$ to a very low value can

also degrade the performance. One possible explanation is that, as $\beta/\lambda \to 0$, the weighting function $F(x, y^i | y^{1,\cdots,K})$ converges to a one-hot distribution, where the response with the highest reward is assigned a weight of 1 and the rest are ignored. This limits learning from a few samples, making it susceptible to simply memorizing the training data [26].

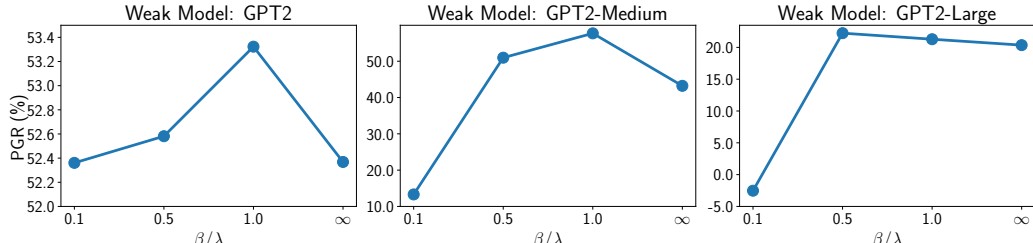

Figure 5: PGR (%) of various $\beta/\lambda$ ratios in EVE's objective.

**Scaling dataset size.** We additionally study the impact of scaling the number of responses $K$ per prompt. Fig. 6 shows the performance of EVE and SFT. EVE demonstrates improved performance as we increase the size of the training dataset (especially as the weak teacher is stronger), while SFT's performance decreases. This can be explained by the fact that as the training data size increases, the strong student also becomes more susceptible to learning the weak teacher's mistakes. In contrast, EVE is designed to avoid this overfitting problem, thus, it can leverage the increased supervision significantly better.

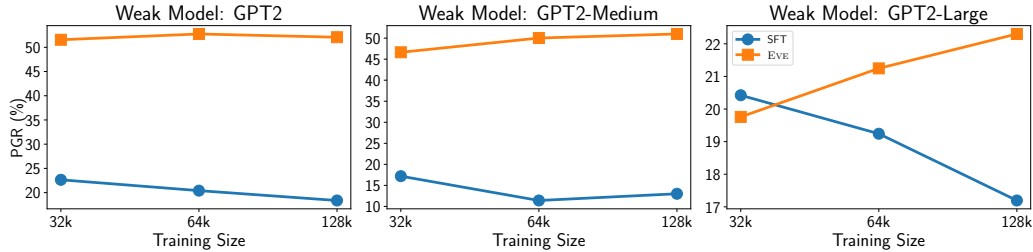

Figure 6: Scaling the training size (32k, 64k and 128k) in EVE and SFT (trained for one epoch). EVE shows notable improvement as the training size increases, while SFT suffers from overfitting.

## 5.2 Instruction Following

**Setup.** We use Qwen2.5-7B as the strong student and Llama-3.2-1B as the weak teacher. The strong reference model $\pi_{\text{base}}^{\text{strong}}$ is initialized from the pre-trained distribution, and the weak model $\pi^{\text{weak}}$ is fine-tuned with DPO on the UltraFeedback dataset [7].

**Evaluation.** We evaluate EVE on two standard instruction-following benchmarks, AlpacaEval 2.0 [9] and IFEval [46]. For AlpacaEval 2.0, we report length-controlled win-rates against `gpt4-turbo`, with `gpt-4o-mini` serving as the judge.

**Baselines.** We evaluate EVE against **SFT, Refinement** and **DPO** - which uses the weak teacher as reward signal to label preference data generated by the strong student. Following prior works [31, 11], we train DPO for 1 epoch with $\beta = 0.05$ as default hyperparameters.

**Results.** We report the results in Fig. 7. EVE consistently outperforms the other W2S approaches across all benchmarks. Interestingly, we find that weak supervision can provide a reliable signal for guiding the strong student, not only in encouraging instruction-following behavior but also helping to filter out non-compliant responses.

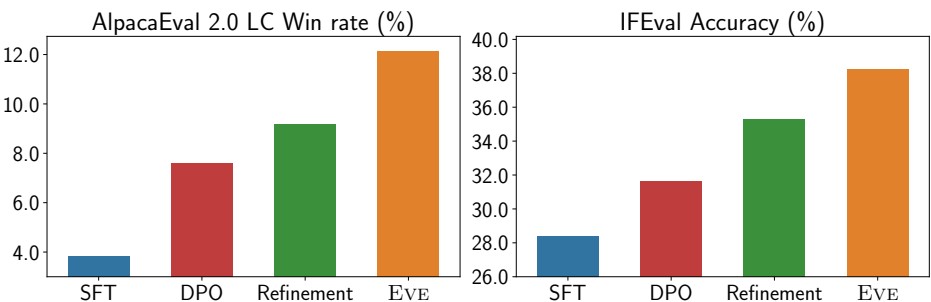

Figure 7: Results on various instruction following benchmarks for various methods.

## 5.3 MLE and Contrastive Learning in W2S Generalization

Fig. 7 also shows the advantages of EVE over contrastive learning approaches (e.g., DPO). Standard RLHF frameworks (e.g., PPO [33] and DPO [30]) can be seen as optimizing the reverse KL, while EVE optimizes the forward KL. As noted in [36], the reverse KL can modify the probability mass more aggressively than forward KL, resulting in a large deviation from $\pi_{\text{ref}}$ to find the peak reward region. Conversely, the forward KL tends to deviate less from its initial distribution towards the peak reward. This might be beneficial in W2S learning due to the following reasons:

**(i) Unreliable learning signal**. Unlike standard RLHF, W2S's feedback is highly unreliable. Finding the peak reward region with the reverse KL can result in performance degradation due to over-optimization as observed in [44] (and re-confirmed in Fig 9 in our Appendix). Importantly, over-optimization can be more severe in W2S generalization as even humans cannot providea reliable signal to avoid these undesirable behaviors [4].

**(ii) Already capable strong student.** Similar to prior works [4, 44, 6], we assume that the strong student is already capable of solving the target tasks. Consequently, we hypothesize that the response region that achieved high rewards should be near the strong student. Therefore, the forward KL, inducing less deviation from the initial distribution, can be seen as an additional implicit regularization and performs better.

## 6 Limitations

We did not experiment with larger language models (> 7B) due to limited computational resources. Given the resource demands of generating data from the strong student, future work will focus on using the strong student for evaluation to eliminate the need for generating strong student data. For example, one direction is to explore extensions to our strategy proposed in Section D.1, which relies on the strong student only for reward evaluation. Tajwar et al. [36]. While our method does introduce additional memory overhead from teacher feedback calculations, this cost is relatively minimal compared to the overall training process of the strong student.

## 7 Conclusion and Discussion

This paper studies the W2S generalization and provides a new theoretical perspective on imitating the weak teacher. We show that imitating the weak teacher is equivalent to maximizing an implicit reward and regularizing the student towards the weak reference policy, which can amplify the bias or mistakes of this supervised fine-tuned weak teacher while not effectively eliciting knowledge from the strong student. Building upon this observation, we propose EVE, which directly optimizes the strong student using an RLHF objective with the "forward KL" regularization towards its latent knowledge of the given task. Extensive empirical results demonstrate that EVE achieves superior performance to existing W2S baselines and effectively mitigates the overfitting problem in W2S generalization.

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
