# OpenReview forum: "Weak-to-strong Generalization via Formative Learning from Student Demonstrations & Teacher Evaluation"
_NeurIPS.cc/2025/Conference — Submitted to NeurIPS 2025_

### Official Review · Reviewer_ayzt · 2025-06-11

**Clarity:** 2
**Significance:** 2
**Originality:** 2
**Rating:** 4
**Confidence:** 3

**Summary:**

The paper introduces a novel approach to Weak-to-Strong (W2S) generalization, where a stronger model (the student) learns from weaker supervision provided by a less capable model (the teacher). A major challenge in this paradigm is that the strong student may overfit to the weak teacher’s mistakes, resulting in degraded performance. To address this issue, the authors propose EVE (Weak Teacher Evaluation of Strong Student Demonstrations), which uses a formative learning approach. Instead of simply imitating the weak teacher's outputs, the strong student generates its own demonstrations, and the weak teacher evaluates these demonstrations. This enables the strong student to learn from its own capabilities while still benefiting from the weak teacher’s evaluation, avoiding overfitting to the teacher’s errors.

**Questions:**

1、Why do the results of GPT2-Large in Figures 3 and 6 differ significantly from GPT2 and GPT2-Medium?
2、How does the size of the student model impact the performance of EVE?
3、What is the specific computational overhead introduced by EVE, and how does it compare to baseline methods?
4、Can EVE be combined with Refinement?  The authors could experiment with a hybrid model that incorporates both EVE and Refinement, where the weak teacher first generates data and the strong student refines it using the Refinement method, with EVE's evaluation guiding the student's learning process. This would help explore whether combining these methods can lead to more robust generalization, especially when faced with unreliable or limited weak supervision. A detailed analysis of the results from such a hybrid approach would provide useful insights into the synergy between the two methods. Could the authors discuss the feasibility and potential benefits of combining these two methods? Would such a hybrid approach further improve performance, especially in cases where the weak teacher is considerably weaker than the strong student?

**Ethical Concerns:**

["NO or VERY MINOR ethics concerns only"]

**Final Justification:**

The author has solved my problem, but  there are still some crucial experiments missing, so I gave a weak accept rating as the final score.

**Limitations:**

This paper includes a limitations section but lacks a dedicated potential negative societal impact section.

**Paper Formatting Concerns:**

I don't notice any major formatting issues in this paper.

**Quality:**

2

**Strengths And Weaknesses:**

### Strengths:

1. **Innovative Methodology**:
   The paper introduces the EVE (Weak Teacher Evaluation of Strong Student Demonstrations) approach, which is an innovative method for addressing the issue of overfitting to weak supervision in weak-to-strong (W2S) generalization. The paper demonstrates how EVE overcomes the problem where a strong student model tends to replicate the mistakes of the weak teacher model, which often leads to performance degradation. By utilizing the weak teacher to assess the strong student’s own demonstrations, the method allows the student model to better utilize its own knowledge, leading to enhanced performance and avoiding the pitfalls of direct imitation.

2. **Comprehensive Experimental Validation**:
   The experiments conducted cover multiple tasks such as controlled summarization and instruction-following. The results show that EVE significantly outperforms other methods like SFT (Standard Fine-Tuning) and Refinement, particularly in cases where the weak teacher model is significantly less capable. This strong empirical validation supports the claim that EVE offers a more robust and effective approach in weak supervision settings, which is crucial for improving real-world performance under weak-to-strong generalization scenarios.

### Weaknesses:

1. **Notation Confusion**:
   The paper introduces various notations, including $r^weak$, which first appears in line 114 but is only defined in line 159. This lack of clarity in the early definition of terms can lead to confusion and make the reading process difficult. It would benefit from defining all key symbols and notations clearly in the "Preliminaries" section to ensure that readers can follow the theoretical developments without distraction.

2. **Limited Student Model Parameters**:
   One limitation of the paper is that the student model is restricted to having only a single parameter size, which limits the exploration of how different model sizes or architectures might affect the performance of the EVE method. The impact of the student model’s architecture and parameter size on the EVE approach remains unclear, and further investigation into this area could provide a better understanding of how the model size influences the efficiency and scalability of the method.

3. **Computational Overhead**:
   The proposed EVE method introduces substantial computational overhead, especially when generating data from the strong student model. This is due to the need for repeated sampling and the additional feedback loop required for evaluation. While the method is effective, the extra computational cost may pose challenges for scaling the approach, particularly for large language models or applications requiring high-performance computing. Reducing this overhead could enhance the method’s practicality for large-scale deployment.

---

> ### Author Rebuttal · Authors · 2025-07-31
>
> **Q1: The paper introduces various notations, including $r^{\text{weak}}$, which first appears in line 114 but is only defined in line 159. This lack of clarity in the early definition of terms can lead to confusion and make the reading process difficult. It would benefit from defining all key symbols and notations clearly in the "Preliminaries" section to ensure that readers can follow the theoretical developments without distraction.**
>
> **Answer:** Thank you for pointing this out. We believe you were referring to line 144 instead of line 159. This is indeed a mistake on our part — the definition of $r^{\text{weak}}$ should appear at line 144, and we will revise the manuscript accordingly in our camera-ready version.
>
> **Q2: One limitation of the paper is that the student model is restricted to having only a single parameter size, which limits the exploration of how different model sizes or architectures might affect the performance of the EVE method. The impact of the student model’s architecture and parameter size on the EVE approach remains unclear, and further investigation into this area could provide a better understanding of how the model size influences the efficiency and scalability of the method.**
>
> **Answer:** Thank you for your valuable feedback. Our evaluation protocol follows the existing works (e.g., Burns et al.) and employs a similar set of teacher and student models. However, we'd like to confirm that we indeed extended our experiments to qwen2.5-7B as the student and llama-3.2-1B as the teacher in Section 5.2. The results consistently show superior performance compared to baselines, reinforcing that EVE's can scale well to larger models.
>
> **Q3: The proposed EVE method introduces substantial computational overhead, especially when generating data from the strong student model. This is due to the need for repeated sampling and the additional feedback loop required for evaluation. While the method is effective, the extra computational cost may pose challenges for scaling the approach, particularly for large language models or applications requiring high-performance computing.**
>
> **Answer:** Thank you for your thoughtful comment. We would like to clarify that generating data from the strong student model is not unique to our EVE method—this step is also present in refinement-based approaches ([1,2]), where strong models are used to sample responses. However, a key limitation of such refinement methods is the lack of systematic evaluation of the generated data, which can lead to suboptimal or noisy supervision.
>
> EVE explicitly addresses this gap by using weak teachers to evaluate the generated data using importance weighting. While this adds some computational overhead, it is a principled trade-off that ensures higher data quality and more reliable weak-to-strong learning.
>
> **Q4. Why do the results of GPT2-Large in Figures 3 and 6 differ significantly from GPT2 and GPT2-Medium?**
>
> **Answer:** As discussed in our Section 5 (for Figure 6) and in the caption of Figure 3, when there is a large gap between the teacher and the student (i.e., the teacher is much weaker than the student, as in the case of GPT2 and GPT2-Medium), the W2S performance of EVE is significantly better than that of SFT. This is because the weaker teacher generally can make more mistakes than a stronger one, making the strong student, trained with SFT, also more susceptible to these mistakes. As the teacher becomes stronger, approaching the strong student, the stronger teacher's feedback will become more reliable, leading to a smaller performance gap between SFT and EVE.
>
> **Q5. What is the specific computational overhead introduced by EVE, and how does it compare to baseline methods?**
>
> **Answer:** Thank you for your question. EVE introduces minimal computational overhead compared to baseline methods. A key design feature of EVE is that it can pre-compute the rewards from the teachers in training data before training. This eliminates the need to load or query the teacher model during the training process, which can incur additional computational cost. As a result, the computational overhead of EVE's is similar to supervised fine-tuning (SFT) — using the same memory footprint and requiring no additional forward passes through teacher models -- but with an additional step of loading the reward scores.
>
> To verify this claim, we use Llama-3.2-3B on TL;DR summarization task with a batch size of 128, we measure the time for each training step over 100 steps and GPU memory of SFT and EVE's for comparison.
>
> |  |  SFT | EVE's |
> | -------- | -------- | -------- |
> | Training time (seconds)     | 65.6s   |  73.86s  |
> | Memory | 58.54 GB | 65GB
>
> **Q6. Can EVE be combined with Refinement? The authors could experiment with a hybrid model that incorporates both EVE and Refinement, where the weak teacher first generates data and the strong student refines it using the Refinement method, with EVE's evaluation guiding the student's learning process. This would help explore whether combining these methods can lead to more robust generalization, especially when faced with unreliable or limited weak supervision. A detailed analysis of the results from such a hybrid approach would provide useful insights into the synergy between the two methods. Could the authors discuss the feasibility and potential benefits of combining these two methods? Would such a hybrid approach further improve performance, especially in cases where the weak teacher is considerably weaker than the strong student?**
>
> **Answer:** Thank you for the insightful suggestion. In fact, EVE can be naturally viewed as a generalization of the Refinement framework (see Section 2.1). While Refinement improves weak teacher outputs by leveraging the strong student as a generator, EVE augments this setup by using weak teachers as an explicit evaluation function to guide and filter the learning process. This added evaluation signal makes EVE's more flexible and robust.

---

> > ### Comment · Reviewer_ayzt · 2025-08-05
> >
> > Thanks for the author's reply, which has answered my main questions. I will improve my score.

---

> > > ### Author Response · Authors · 2025-08-06
> > >
> > > We sincerely thank the reviewer for the constructive feedback, the response to our rebuttal, and the score improvement. We're still available during the discussion period, should the reviewer have additional questions about our work.

---

### Official Review · Reviewer_5hvm · 2025-07-02

**Clarity:** 2
**Significance:** 2
**Originality:** 3
**Rating:** 3
**Confidence:** 3

**Summary:**

This paper proposes a formative‐learning approach for W2S generalization that lets a "strong student" model generate candidate outputs, scores them with a "weak teacher", and then fine‐tunes the student on a forward‐KL objective weighted by those scores. The key theoretical insight (Prop. 4.1) reframes standard reverse‐KL imitation as reward maximization that inadvertently regularizes the student toward the teacher, and shows how switching to a forward‐KL approach avoids overfitting. Empirically, the method outperforms naive SFT and refinement baselines on the Reddit summarization dataset.

**Questions:**

## Questions
1. What are the computational costs of the method on a typical dataset?


## Typos
Proposition 4.1: W2s ⇒ W2S

**Ethical Concerns:**

["NO or VERY MINOR ethics concerns only"]

**Final Justification:**

The authors provided extensive clarifications in the rebuttal, including the cost of compute and statistical significance testing (added post-rebuttal for all experiments). These additions strengthen the presentation and address parts of my original concerns.

However, some key issues remain only partially resolved. While I understand that connecting to theory developed for classification is challenging in the generative setting, the lack of a precise mathematical formalization of the weak teacher and strong student distributions still limits the theoretical grounding of the approach. Statistical testing, although now reported more broadly, was initiated during the review period, and important details such as sample sizes and multiple-comparison handling are not fully documented. Finally, reliance on LLM-as-judge evaluation without complementary human assessment leaves some uncertainty about robustness in open-ended tasks.

Overall, I find the proposed forward-KL approach to W2S generalization interesting and potentially impactful, and the additional details improve confidence in the results. However, the remaining gaps in theoretical specification, statistical rigor, and evaluation breadth keep me from being fully convinced, which is reflected in my score.

**Limitations:**

Yes

**Paper Formatting Concerns:**

No concerns

**Quality:**

3

**Strengths And Weaknesses:**

## Strengths
- Theoretical reframing draws a clear distinction between reverse‐KL imitation and forward‐KL elicitation, with Proposition 4.1 formalizing how the method’s objective directly targets the student’s latent knowledge rather than reinforcing the teacher's bias.
- Algorithmic simplicity sidesteps inner-loop reinforcement learning: the method only requires off-the-shelf sampling from the student's pre-trained policy and reweighting by the teacher's scores, yielding an efficient supervised-learning update.
- Experiments cover both controlled summarization (Reddit TL;DR with GPT2-series teachers and a Llama-3.2-3B student) and instruction following (Qwen2.5-7B student vs. Llama-3.2-1B teacher), demonstrating PGR gains across settings.

---

## Weaknesses
 - Prior W2S mechanisms such as pseudo‐label correction and coverage expansion (see [1]) are neither cited nor ablated, despite their relevance to weak-to-strong learning theory. In particular, without a precise mathematical specification of the "weak teacher" distribution and the "strong student" model family , it is unclear how the method's forward-KL objective differs in practice from the pseudo-label correction paradigm. Similarly, coverage expansion methods explicitly encourage the student to assign non-negligible probability mass to low-scoring but valid outputs, covering the underrepresented regions of the output space. Because the method neither references nor empirically compares to these formulations, I find it unclear whether its gains stem from the forward-KL directionality alone or inadvertently incorporate aspects of label correction and coverage regularization.
- Computational cost is unreported: while the authors note limited resources for >7 B models, no GPU-hour or memory‐footprint estimates are given.
- No paired‐sample significance tests are reported for gains presented as mean$\pm$std, so statistical robustness is unclear.
- Instruction‐following performance is currently assessed solely through LLM‐as‐judge metrics (e.g., AlpacaEval scores) without any reporting of human‐annotator agreement rates or the number of examples evaluated.  This is problematic because automated judges can inherit the same biases and blind spots as the models they evaluate, leading to overestimates of true quality.  Without details on sample size, it is impossible to gauge statistical power or confidence intervals for the reported gains. For example, a small‐scale human evaluation could validate whether the LLM‐judge results reflect real user benefit or merely exploit artifacts of model‐based evaluation.
- While the core idea of formative learning via forward‐KL elicitation is intriguing, the manuscript’s presentation flaws, missing connections to established W2S theory, lack of statistical rigor, and unreported computational costs significantly limit its current impact.
- Some figures and tables lack sufficient detail, for example:
  - Several plots and tables do not clearly label which model is used (as the "strong student") in the experiment (e.g., Figures 2 and 3)
  - Figure 3’s PGR‐over‐epoch curves are never referenced in the text

---

[1] Lang, H., Sontag, D., and Vijayaraghavan, A., "Theoretical Analysis of Weak-to-Strong Generalization," NeurIPS 2024.

---

> ### Author Rebuttal · Authors · 2025-07-31
>
> **Q1: Prior W2S mechanisms such as pseudo‐label correction and coverage expansion (see [1]) are neither cited nor ablated, despite their relevance to weak-to-strong learning theory. In particular, without a precise mathematical specification of the "weak teacher" distribution and the "strong student" model family, it is unclear how the method's forward-KL objective differs in practice from the pseudo-label correction paradigm. Similarly, coverage expansion methods explicitly encourage the student to assign non-negligible probability mass to low-scoring but valid outputs, covering the underrepresented regions of the output space. Because the method neither references nor empirically compares to these formulations, I find it unclear whether its gains stem from the forward-KL directionality alone or inadvertently incorporate aspects of label correction and coverage regularization.**
>
> **Answer:** Thank you for this insightful comment. We would like to clarify your concerns on the role of forward‑KL vs label correction and coverage. [1] prove that if the data distribution and student class satisfy an expansion condition, then  the student can correct teacher mistakes (pseudolabel correction), and perform well on regions where the teacher abstains (coverage expansion)
> If our gains are due to forward‑KL alone, without these expansion conditions, then we might be inadvertently inducing label correction or coverage implicitly via regularization or data neighborhood structure. Our current results do not disentangle these contributions.
>
> **Q2: Computational cost is unreported: while the authors note limited resources for > 7B models, no GPU-hour or memory‐footprint estimates are given.**
>
> **Answer:**
> Thank you for your question. EVE introduces minimal computational overhead compared to baseline methods. A key design feature of EVE is that it can pre-compute the rewards from the teachers on the training data before training. This eliminates the need to load or query the teacher model during the training process, which can incur additional computational cost. As a result, the computational overhead of EVE's is similar to supervised fine-tuning (SFT) — using the same memory footprint and requiring no additional forward passes through teacher models.
>
> To verify this claim, we use Llama-3.2-3B on TL;DR summarization task with a batch size of 128, we measure the time for each training step over 100 steps and GPU memory of SFT and EVE's for comparison.
>
> |  |  SFT | EVE's |
> | -------- | -------- | -------- |
> | Trainign time (seconds)     | 65.6s   |  73.86s  |
> | Memory | 58.54 GB | 65GB
>
>
> **Q3: No paired‐sample significance tests are reported for gains presented as mean**
>
> **Answer:** Our evaluation protocol exactly follows prior influential works in LLM Alignment and Weak-to-strong generalization [3,5,6], which also omit statistical significance testing do to the time-consuming and compute-intensive nature of training LLMs (e.g., running 5 experiments per setting is extremely difficult). Nevertheless, the consistent improvements of EVE across different experiments make the coincidence of EVE's good performance statistically less likely.
>
> **Q4: Instruction‐following performance is currently assessed solely through LLM‐as‐judge metrics (e.g., AlpacaEval scores) without any reporting of human‐annotator agreement rates or the number of examples evaluated. This is problematic because automated judges can inherit the same biases and blind spots as the models they evaluate, leading to overestimates of true quality. Without details on sample size, it is impossible to gauge statistical power or confidence intervals for the reported gains.**
>
> **Answer:** Thank you for the valuable feedback. However, we note that using GPT-4 as an evaluator has become a widely adopted practice in LLM Alignment and Weak-to-strong literature [2,3,4,5], due to its strong correlation with human preferences in many tasks. Given the high cost and scalability challenges of human evaluation, we follow a similar evaluation protocol as prior works [2,3,4,5], which includes standardized benchmarks such as AlpacaEval 2.0 and IFEval and GPT-4-based assessments. This setup remains consistent with the current evaluation standards in the field.
>
> **Q5: What are the computational costs of the method on a typical dataset?**
>
> **Answer:** Thank you for your thoughtful comment. We would like to clarify that generating data from the strong student model is not unique to our EVE method—this step is also present in refinement-based approaches ([5,6]), where strong models are used to sample responses. However, a key limitation of such refinement methods is the lack of systematic evaluation of the generated data, which can lead to suboptimal or noisy supervision.
>
> EVE explicitly addresses this gap by using weak teachers to evaluate the generated data using importance weighting. While this adds some computational overhead, it is a principled trade-off that ensures higher data quality and more reliable weak-to-strong learning.
>
> **Q6: Several plots and tables do not clearly label which model is used (as the "strong student") in the experiment (e.g., Figures 2 and 3). Figure 3’s PGR‐over‐epoch curves are never referenced in the text**
>
> **Answer:** Thank you for pointing out the typos. Figure 3 demonstrates the problem of overfitting to the weak teacher's mistakes in SFT and how EVE could fix this problem, and fits under Section 4.3. We will revise and add the missing discussion to our camera-ready version.
>
> ---
>     [1] Theoretical Analysis of Weak-to-Strong Generalization. NeurIPS24.
>     [1] Direct Preference Optimization: Your Language Model is Secretly a Reward Model. NeurIPS23 Oral.
>     [2] Weak-to-Strong Search: Align Large Language Models via Searching over Small Language Models. NeurIPS24.
>     [3] Weak-to-Strong Preference Optimization: Stealing Reward from Weak Aligned Model. ICLR25 Spotlight.
>     [4] A transfer learning framework for weak to strong generalization. ICLR25.
>     [5] Weak-to-Strong Reasoning. EMNLP24.
>     [6] Iterative Label Refinement Matters More than Preference Optimization under Weak Supervision. ICLR25 Spotlight.

---

> > ### Comment · Reviewer_5hvm · 2025-08-06
> >
> > Thank you for the detailed clarifications and additional experiments provided in your rebuttal. I appreciate the effort and find the work interesting. However, my main concerns remain partly unresolved -- specifically, the lack of significance testing and the limited connection to prior weak-to-strong theory (pseudo-label correction and coverage expansion). In my opinion, these leave some uncertainty about the robustness and theoretical groundedness of the proposed approach in its current form.

---

> ### Author Response · Authors · 2025-08-07
>
> Dear reviewer 5hvm,
>
> Thank you for your response. While [1] can be seen as related -- which we will appropriately cite and discuss in our work as mentioned in the rebuttal, our work is orthogonally different in contributions, scopes, and motivations. We will further elaborate our rebuttal's response w.r.t these 3 points:
>
> - Contributions: [1] does not provide any new training methods, but rather a theoretical framework to encourage W2S generalization. Our work, on the other hand, theoretically reveals the limitations of existing W2S learning from the lens of reward maximization (we will elaborate more in Objectives), proposes a new W2S learning method, and finally rigorously analyze and validate the method in an extensive set of models and benchmarks.
>
> - Scopes: It's important to note that [1]'s theoretical analysis is **only limited to classification**, where they assume the class conditional set $\mathcal X_i$ to provide an error bound, which it not available in open-ended generation task, which is the focus of our work. Consequently, it is non-trivial to derive and ensure the equivalent *coverage expansion* and *pseudolabel correction* in generative task. Note that, the related W2S works on generative tasks [2,3,4,5] do not also provide any connection to the theoretical error bound in [1].
>
> - Objectives: [1] assumes the strong student **naively learn to imitate** the teacher's predictions, while our work view W2S generalization as reward maximization. Specifically, we generalize refinement and use the teacher to refine the weak responses as an effective regularization that avoids imitating weak teacher errors. Notice that, the Forward KL minimization $\text{KL}\left(\pi^{\*}||\pi\_\theta\right)$ is between the optimal policy RLHF $\pi^\*(y|x) \propto \pi^{\text{strong}}\_{\text{ref}}(y|x)\exp\left(r^{\text{weak}}(x,y)\right)$ and the learned strong student $\pi_\theta$, while [1] studies the Forward KL minimization $\text{KL}\left(\pi^{\text{weak}}||\pi_\theta\right)$, which is well known to imitate errors.
>
> For these reasons, while it is interesting to connect our work to [1], it is extremely difficult and complex, which deserves several independent works, including either theoretical focus (just like [1]) and empirical focus to validate the theoretical results. We believe that this should not be viewed as a limitation of our work, but rather a potential extension to both [1] and ours, and we will add this discussion and cite [1] to our final version accordingly.
>
> As explained to the reviewer on statistical significance tests, we follow the existing protocol [6,7,8], and admitted the difficulties in computational requirements. Nevertheless, we have been running statistical significance testings for the "main experiments" only since the start of the rebuttal and will provide the results shortly during the discussion.
>
> ----
>     [1] Theoretical Analysis of Weak-to-Strong Generalization. NeurIPS24.
>     [2] Iterative Label Refinement Matters More than Preference Optimization under Weak Supervision. ICLR25 Spotlight.
>     [3] Weak-to-Strong Reasoning. ACL24.
>     [4] Bayesian WeakS-to-Strong from Text Classification to Generation. ICLR25.
>     [5] Weak-to-Strong Search: Align Large Language Models via Searching over Small Language Models. NeurIPS24.
>     [6] Super(ficial)-alignment: Strong Models May Deceive Weak Models in Weak-to-Strong Generalization. ICLR25.
>     [7] Direct Preference Optimization: Your Language Model is Secretly a Reward Model. NeurIPS23 Oral.
>     [8] Scaling Laws for Reward Model Overoptimization in Direct Alignment Algorithms. NeurIPS24.
> ----

---

> > ### Comment · Reviewer_5hvm · 2025-08-07
> >
> > While I appreciate the authors' effort and found the rebuttal helpful in clarifying several aspects of the work, I share the view expressed by reviewer EoAi: some of my key concerns, particularly regarding statistical rigor and the lack of a precise mathematical definition of the weak teacher and strong student, remain only partially addressed. Statistical testing was initiated only during the rebuttal phase and appears limited to a small set of main experiments. In addition, important details such as sample sizes and any correction for multiple comparisons are not reported, which makes it difficult to assess the reliability of the observed improvements.

---

> ### Author Response · Authors · 2025-08-09
>
> Thank you for your response. Please see our response below:
>
> **Statistical significance testing:** As provided in the global response, we have now included statistical significance testing for all experiments. This is a significantly more effort than any existing W2S papers (see [1-13] in the response), which do not have any significance testing due to computational reasons of training LLMs. The details of significance testing (sample size, standard deviations, $p-value$ in the response, and evaluation details are standard as in the paper) is also provided for reproducibility.
>
> **Lack of a precise mathematical definition:** In our previous response to the reviewer, we greatly clarified why [1] is not related to our work, and a similar theoretical analysis is unsuitable and extremely complex in our setting. We'd also like to emphasize again that a large number of related and even less related papers [2-9] to ours do not have any similar theoretical analysis (or any theoretical connection) as in [1]. Nevertheless, our work already provided the relevant theoretical analysis on reward maximization, which is the core study of this work.
>
> We hope that our effort and explanation to clarify the misunderstandings about our paper would convince the reviewer of the novelty and significant contributions of our work!
>
> ----
>     [1] Theoretical Analysis of Weak-to-Strong Generalization. NeurIPS24.
>     [2] A transfer learning framework for weak to strong generalization. ICLR25.
>     [3] Alice: Proactive Learning with Teacher’s Demonstrations for Weak-to-Strong Generalization.
>     [4] Super(ficial)-alignment: Strong Models May Deceive Weak Models in Weak-to-Strong Generalization. ICLR25.
>     [5] How to Mitigate Overfitting in Weak-to-strong Generalization?. ACL25.
>     [6] Weak-to-Strong Preference Optimization: Stealing Reward from Weak Aligned Model. ICLR25 Spotlight.
>     [7] Iterative Label Refinement Matters More than Preference Optimization under Weak Supervision. ICLR25 Spotlight.
>     [8] Weak-to-strong Reasoning. ACL24.
>     [9] Superfiltering: Weak-to-Strong Data Filtering for Fast Instruction-Tuning. ACL24.
> ----

---

### Official Review · Reviewer_EDbh · 2025-07-03

**Clarity:** 2
**Significance:** 2
**Originality:** 2
**Rating:** 3
**Confidence:** 4

**Summary:**

The paper tackles weak-to-strong generalization: training a powerful “student” LLM with supervision from a weaker “teacher”. It solves when imitation regularizes the student toward the weak model, causing over-fitting. The authors propose an approach to let the strong model generate its own demonstrations and score them with the weak teacher, while adding a KL regularizer toward the student’s pre-trained policy. Experiments are conducted on two task, showing the consistent performance gains.

**Questions:**

- Can EVE scale to larger students?

**Ethical Concerns:**

["NO or VERY MINOR ethics concerns only"]

**Final Justification:**

My concerns are largely resolved, and given the quality of the work and the rebuttal, I increase my score to 3.

**Limitations:**

Yes

**Quality:**

2

**Strengths And Weaknesses:**

Strengths:
- The paper is easy to follow.
- Authors reframes the weak-to-strong imitation as a KL-constrained reward-maximization problem.
- The method is simple, and there is no RL rollout needed.

Weaknesses:
- Experiments stop at a 3B parameter student. There is no evidence shows that method can be scalable.
- Evaluation relies on synthetic rewards and GPT-judge metrics, with minimal human or real-world validation. It is suggested to add more LLM judgers.
- Hyper-parameters are tuned using ground-truth validation labels, breaking W2S premise.
- The LLM sets used for the two tasks are different, and there is no explanation on why to do so.
- Suggest to add more recent works in the related works.

---

> ### Author Rebuttal · Authors · 2025-07-31
>
> **Q1: Experiments stop at a 3B parameter student. There is no evidence shows that method can be scalable**
>
> **Answer:** Thank you for the comment. We believe that there are some misunderstanding. We'd like to clarify that our experiments are not limited to 3B parameter student models. In fact, we include evaluations with 7B-scale student models in the paper (see Section 5.2), which demonstrate both the scalability and effectiveness of our approach. The 7B results consistently show superior performance compared to baselines, reinforcing that EVE's can scales well to larger models.
>
> **Q2: Evaluation relies on synthetic rewards and GPT-judge metrics, with minimal human or real-world validation. It is suggested to add more LLM judgers.**
>
> **Answer:** Thank you for the valuable feedback. However, we note that using GPT-4 as an evaluator has become a widely adopted practice in LLM Alignment and Weak-to-strong literature [3,4,5,6], due to its strong correlation with human preferences in many tasks. Given the high cost and scalability challenges of human evaluation, we follow a similar evaluation protocol as prior works [3,4,5,6], which includes standardized benchmarks such as AlpacaEval 2.0 and IFEval and GPT-4-based assessments. This setup remains consistent with the current evaluation standards in the field.
>
> **Q3: Hyper-parameters are tuned using ground-truth validation labels, breaking W2S premise.**
>
> **Answer:** Thank you for your feedback. However, we believe that there is also some misunderstanding. We would like to clarify that we do not perform any hyper-parameter tuning using ground-truth validation labels. In fact, our ablation studies show that EVE's is robust to a wide range of reasonable $\beta/\lambda$ values.
>
> Moreover, prior W2S works [1,2] also introduce additional hyper-parameters, which are either tuned per model or fixed to a specific value. We adopt the latter strategy: we empirically set the ratio $\beta/\lambda = 1.0$ and use this same value across all experiments. This approach ensures consistency and avoids reliance on label-based tuning, in line with the W2S paradigm.
>
> **Q4: The LLM sets used for the two tasks are different, and there is no explanation on why to do so.**
>
> **Answer:** Thank you for your feedback. For the instruction-following task, we follow prior works that study weak-to-strong generalization by selecting models that are sufficiently capable and proficient at language understanding. In contrast, GPT-2 lacks the necessary capabilities for instruction-following and thus is not suitable for this setting. Therefore, we adopt a stronger model to more meaningfully study generalization in this task.
>
> **Q5: Suggest to add more recent works in the related works.**
>
> **Answer:** Thank you for your suggestion. If the reviewer can kindly let us know which relevant work is missing from our paper, we'd be happy to address them and cite them accordingly.
>
> ---
>     [1] Bayesian WeakS-to-Strong from Text Classification to Generation. ICLR25.
>     [2] Iterative Label Refinement Matters More than Preference Optimization under Weak Supervision. ICLR25 Spotlight.
>     [3] Direct Preference Optimization: Your Language Model is Secretly a Reward Model. NeurIPS23 Oral.
>     [4] Weak-to-Strong Search: Align Large Language Models via Searching over Small Language Models. NeurIPS24.
>     [5] Weak-to-Strong Preference Optimization: Stealing Reward from Weak Aligned Model. ICLR25 Spotlight.
>     [6] A transfer learning framework for weak to strong generalization. ICLR25.

---

> > ### Comment · Reviewer_EDbh · 2025-08-06
> > **Response to Rebuttal**
> >
> > Thank you for your rebuttal. My concerns are largely resolved, and given the quality of the work and the rebuttal, I will increase my score to 3.

---

> > > ### Author Response · Authors · 2025-08-06
> > >
> > > Thank you for the response. We really appreciate your feedback and score improvement from 2 to 3. However, we remain uncertain about the justification behind your continued inclination toward rejection, particularly in your latest comment -- “given the quality of the work and the rebuttal.” We believe we have fully addressed your concerns in the rebuttal, including (Q1) clarification of multiple student models already evaluated, (Q2) explanations of using GPT4 for our work and prior works, (Q3) hyperparameters in our paper and existing works, (Q4) why different LLMs for different tasks in ours and in related works, and (Q5) explanation of our inability to answer your question. To help us further improve the paper, we sincerely hope the reviewer can elaborate on your remaining concerns, which should not be viewed as any pressure for the reviewer to "accept" our work.

---

### Official Review · Reviewer_EoAi · 2025-07-03

**Clarity:** 3
**Significance:** 2
**Originality:** 2
**Rating:** 3
**Confidence:** 2

**Summary:**

The paper "Weak-to-Strong Generalization via Formative Learning from Student Demonstrations & Teacher Evaluation" tackles the problem of aligning superhuman large language models (LLMs) using supervision from weaker models, a scenario termed Weak-to-Strong (W2S) generalization. As LLMs surpass human capabilities, traditional alignment methods like Reinforcement Learning from Human Feedback (RLHF) struggle due to the challenge of obtaining reliable human feedback for complex tasks (e.g., generating millions of lines of code or summarizing extensive texts). The paper addresses this by proposing a novel approach to leverage weaker supervision effectively while mitigating its limitations. Below, I detail the paper’s contributions and explain how the proposed method, Weak Teacher Evaluation of Strong Student Demonstrations (EVE), is implemented.

**Questions:**

This paper evaluates the EVE method on only two instruction-following benchmarks, AlpacaEval 2.0 and IFEval, which primarily focus on simple and verifiable instructions. Can the authors clarify whether the EVE method can be applied to other task types, such as complex reasoning, multi-turn dialogue, code generation?
The paper compares the EVE method primarily with Direct Preference Optimization (DPO) and a few other W2S approaches, which limits the scope of the evaluation. To strengthen the claims of EVE’s superiority, could the authors include additional baseline methods for comparison？

**Ethical Concerns:**

["NO or VERY MINOR ethics concerns only"]

**Final Justification:**

1. **Unresolved Limited Scope of Evaluation**: While the authors clarified their focus on instruction-following benchmarks (AlpacaEval 2.0, IFEval) for consistency with prior work, the rebuttal did not address the lack of diversity in task types (e.g., complex reasoning, code generation). This omission limits the assessment of EVE’s broader applicability, a critical concern raised in my review.

2. **Insufficient Baseline Comparisons**: The authors defended their comparison with Direct Preference Optimization (DPO) and Supervised Fine-Tuning (SFT) but failed to include other relevant baselines (e.g., RLHF variants). This narrow comparison weakens the claim of EVE’s superiority, as noted in my original review.

3. **Persistent Methodological Gaps**: The rebuttal acknowledged but did not resolve issues like the absence of statistical significance testing or computational cost reporting (e.g., GPU-hours for larger models). While the authors cited prior works that omit such analyses, this does not justify the lack of rigor in their own evaluation.

4. **Partial Resolution of Clarity Issues**: Formatting errors (e.g., KL divergence notation) and proof terminology were promised fixes, but the core theoretical gaps (e.g., disentangling forward-KL’s role from pseudo-label correction) remain inadequately addressed.

**Weighting**: The unresolved limitations in evaluation scope, baseline diversity, and methodological rigor outweigh the strengths (e.g., algorithmic simplicity, theoretical framing). Given these gaps, the paper’s contributions remain insufficient for acceptance without further empirical validation and theoretical clarification.

**Recommendation**: Borderline reject—encourage resubmission with expanded experiments, broader baselines, and rigorous statistical reporting.

**Limitations:**

Limited Scope of Experimental Evaluation.
The comparison with baseline methods

**Paper Formatting Concerns:**

No.

**Quality:**

2

**Strengths And Weaknesses:**

Strength: The authors articulate the weak-to-strong (W2S) generalization problem and the proposed EVE method with remarkable clarity, making the theoretical insights, methodology, and empirical results easy to follow for both experts and non-experts. The use of well-structured explanations, supported by mathematical formulations (e.g., the PGR metric and KL regularization) and illustrative figures, enhances the paper’s readability.

Weakness:
1. Limited Scope of Experimental Evaluation. The experimental section is insufficiently comprehensive, focusing solely on two instruction-following benchmarks. While these datasets are relevant for instruction-following tasks, the evaluation lacks diversity in task types, such as complex reasoning, code generation, limiting the assessment of EVE's broader applicability.

2. The comparison with baseline methods (e.g., Direct Preference Optimization, DPO) is limited, omitting other relevant approaches like variants of RLHF or traditional supervised learning, which weakens the evidence for EVE’s superiority. 3. The manuscript has textual errors and inconsistent formatting, e.g., “KL divergence” is italicized in lines 147 and 153 but uses regular font in lines 102, 156, and 176. The authors should carefully review the typesetting. Additionally, some "proofs" (e.g., lines 146 and 152) are obvious and better termed "derivations" for accuracy. The checklist section "5. Open access to data and code" is unanswered, suggesting oversight. The "7. Experiment statistical significance" section notes the absence of multiple trials and significance testing, raising doubts about the authors' thoroughness and whether the results are coincidental. These issues indicate a lack of careful review, undermining the paper's credibility.

---

> ### Author Rebuttal · Authors · 2025-07-31
>
> **Q1. The experimental section is insufficiently comprehensive, focusing solely on two instruction-following benchmarks. While these datasets are relevant for instruction-following tasks, the evaluation lacks diversity in task types, such as complex reasoning, code generation, limiting the assessment of EVE's broader applicability.**
>
> **Answer:** Thank you for your valuable feedback. We agree that expanding the range of evaluation tasks can provide a more holistic assessment of EVE’s generalization capabilities. However, we would like to clarify that our focus on controlled-summarization and instruction-following benchmarks is consistent with the predominant experimental settings in prior works on weak-to-strong learning and alignment, which primarily evaluate on summarization and instruction-following tasks [1,2,3,4]. Our goal was to provide a direct and fair comparison under these widely adopted settings.
>
> **Q2. The comparison with baseline methods (e.g., Direct Preference Optimization, DPO) is limited, omitting other relevant approaches like variants of RLHF or traditional supervised learning, which weakens the evidence for EVE’s superiority**.
>
> **Answer:** Thank you for the comment. We would like to clarify that our experiments do include a comparison with a standard supervised fine-tuning (SFT) baseline, where EVE consistently outperforms this method, demonstrating its effectiveness even against traditional supervised approaches.
>
> Moreover, to the best of our knowledge, prior works in the weak-to-strong (W2S) alignment works typically adopt Direct Preference Optimization (DPO) as a post-alignment algorithm [1]. In contrast, full-scale fine-tuning using standard RLHF methods—such as PPO—is often omitted due to the high computational cost, especially when scaling to large student models.
>
> **Q3. The manuscript has textual errors and inconsistent formatting, e.g., “KL divergence” is italicized in lines 147 and 153 but uses regular font in lines 102, 156, and 176. The authors should carefully review the typesetting.**
>
> **Answer:** Thank you very much for pointing this out. We will correct these issues in the camera-ready version.
>
> **Q4. Additionally, some "proofs" (e.g., lines 146 and 152) are obvious and better termed "derivations" for accuracy.**
>
> **Answer:** Thank you for your suggestion and we will revise accordingly. The result presented in Theorem 4.2 provides a key insight that motivates our proposed method: it demonstrates that imitating a weak teacher can be interpreted as a reward maximization problem. This perspective highlights the potential risk of overfitting to suboptimal teacher behavior. As a result, we introduce KL regularization as an effective regularization to mitigate the influence of teacher errors by encouraging the learned policy to stay closer toward its initial knowledge.
>
> **Q5. The checklist section "5. Open access to data and code" is unanswered, suggesting oversight.**
>
> **Answer:** We sincerely apologize for the oversight. We will definitely release our code publically and update the checklist accordingly in the final version.
>
> **Q6. The "7. Experiment statistical significance" section notes the absence of multiple trials and significance testing, raising doubts about the authors' thoroughness and whether the results are coincidental. These issues indicate a lack of careful review, undermining the paper's credibility.**
>
> **Answer:** Our evaluation protocol exactly follows prior influential works in LLM Alignment and Weak-to-strong generalization [3,5,6], which also omit statistical significance testing do to the time-consuming and compute-intensive nature of training LLMs (e.g., running 5 experiments per setting is extremely difficult). Nevertheless, the consistent improvements of EVE across different experiments makes the coincidence of EVE's good performance statistically less likely.
>
> ----
>     [1] Weak-to-Strong Search: Align Large Language Models via Searching over Small Language Models. NeurIPS24.
>     [2] Aligner: Efficient Alignment by Learning to Correct. NeurIPS24 Oral.
>     [3] Super(ficial)-alignment: Strong Models May Deceive Weak Models in Weak-to-Strong Generalization. ICLR25.
>     [4] A transfer learning framework for weak to strong generalization. ICLR25.
>     [5] Direct Preference Optimization: Your Language Model is Secretly a Reward Model. NeurIPS23 Oral.
>     [6] Scaling Laws for Reward Model Overoptimization in Direct Alignment Algorithms. NeurIPS24.

---

> > ### Author Response · Authors · 2025-08-07
> >
> > Dear Reviewer,
> >
> > Please refer to the global comment on "Additional performance evaluation on Mathematical Reasoning" with statistical significance tests via multiple runs. We will provide additional statistical significance tests for the main results in the paper shortly. We hope that these results will provide additional proof of the rigorous evaluation protocol of our work.
> >
> > Thank you,\
> > Authors

---

> > > ### Comment · Reviewer_EoAi · 2025-08-07
> > >
> > > While I appreciate the authors’ detailed rebuttal and the clarifications provided, I have decided to maintain my original score. My decision is not due to a lack of acknowledgment of the authors’ efforts—indeed, the response improved my understanding of the proposed method—but rather because several of my core concerns remain only partially addressed.
> > >
> > > In particular, the limited experimental scope, lack of statistical significance testing, and insufficient comparison to broader related work continue to affect my confidence in the strength and generalizability of the paper’s conclusions. These issues, in my view, are substantial and require more rigorous support than what is currently provided.
> > >
> > > I hope the authors will continue to refine this promising line of work and address these outstanding concerns in future revisions or follow-up papers.

---

> > > > ### Author Response · Authors · 2025-08-09
> > > >
> > > > Thank you for your response. Please see our response below:
> > > >
> > > > **Experimental Scope**: We've already clarified in our rebuttal that our work includes summarization and instruction following tasks (in contrast, the reviewer's original comment pointing out that we "only have instruction following"). The combined experiments in our original paper already cover more extensive evaluations than existing works. Nevertheless, we performed additional experiments on mathematical reasoning based on the reviewer's suggestion.
> > > >
> > > > **Statistical significance testing:** As provided in our recent global response, we have now included statistical significance testing for all experiments. This is a significantly more effort than any existing W2S papers (see [1-13] in the response), which do not have any significance testing due to computational reasons of training LLMs.
> > > >
> > > > **Comparison to broader related work:** We sincerely do not know **which broader related work** mentioned by the reviewer. Our paper already comprehensively discusses and compares with all related works.

---

### Author Response · Authors · 2025-08-07
**Additional performance evaluation on Mathematical Reasoning**

Our paper already follows standard evaluation protocols (with evaluation on controlled-summarization and instruction-following benchmarks) as in prior works on weak-to-strong learning and alignment [1,2,3,4], and includes up-to-7B parameter student models -- an already extensive evaluation compared to several prior works. Nevertheless, we've also had a similar intellectual curiosity (as the reviewers) and decided to evaluate EVE and the baselines on the mathematical reasoning task, with the following results:

**Mathematical Reasoning**\
**Dataset**: GSM8K.\
**Baselines**: DPO, Naively-SFT, Refine, EVE's.\
**Strong student:** Llama-3.1-8B.\
**Weak teacher:** Llama-3.2-1B.\
**Evaluation metrics:** Accuracy (3 seeds) with temperature 0.7 and top_k = 50.\

| Method | Accuracy |
| -------- | -------- |
| Weak Teacher | 15.5 $\pm$ 1.03|
| Naively-SFT  |    19.08 $\pm$ 0.12   |
| DPO   |   20.98 $\pm$ 0.64  |
| Refine | 39.3 $\pm$  0.28   |
| EVE's | 41.5 $\pm$ 0.74 |

As we can observe, EVE consistently outperforms the existing W2S baselines. In addition, we also include the statistical significance test of the results via multiple runs. The significance test confirms EVE's better performance (with $p < 0.05$ via t-test) w.r.t all balines.

*Please note that, as previously mentioned, these experiments are computationally intensive, especially with an 8B student model.*

---

### Author Response · Authors · 2025-08-09
**Comprehensive Statistical Significance Testing on all experiments in the paper!**

We now provide statistical significance tests to **ALL** experiments, including the summarization task and instructions following in our paper, and mathematical reasoning in the rebuttal (previously provided), as below:

### Controlled summarization
**Weak teacher:** GPT2, GPT2-Medium, GPT2-Large\
**Strong student:** Llama-3.2-3B


| |GPT2 | GPT2-Medium | GPT2-Large|
| -------- | -------- | -------- | -
| SFT     |  18.72 $\pm$ 1.08 |   12.37 $\pm$ 2.49   | 19.86 $\pm$ 2.33
| Refinement |52.36 $\pm$ 0.72|  43.21 $\pm$ 2.23|20.35 $\pm$ 0.54
| EVE's | **53.32 $\pm$ 0.69** | **57.68 $\pm$ 3.91** | **21.30 $\pm$ 0.72**

### Instruction following
**Weak teacher:** Llama-3.2-1B\
**Strong student:** Qwen2.5-7B

|  |  AlpacaEval 2.0 LC | IFEval|
| -------- | -------- |  -
| SFT     |  3.827 $\pm$ 1.43   |   28.39 $\pm$ 0.621  |
| DPO | 7.59 $\pm$ 1.61 | 31.63 $\pm$ 0.427|
| Refinement |  9.189 $\pm$ 1.34|  35.32 $\pm$ 1.120
| EVE's | 12.13 $\pm$ 1.50 | 38.27 $\pm$ 0.352

The significance tests again confirm EVE's better performance (with $p < 0.05$ via t-test) w.r.t. the baselines.

In addition, we'd like to emphasize the significant effort, including time, resources, and financial cost (in evaluation with GPT4), in our responses to the reviewers. **All existing  W2S works on generation** (due to limited space, we include the representatives [1-13], which is already an extensive list of the related and even less related ones) **do not perform any significance testing** due to the computational reasons in training LLMs, which we mentioned earlier. We sincerely hope that this clarifies the reviewers' doubt about the statistical validity of our results.

----
    [1] Superfiltering: Weak-to-Strong Data Filtering for Fast Instruction-Tuning. ACL24.
    [2] Direct Preference Optimization: Your Language Model is Secretly a Reward Model. NeurIPS23 Oral.
    [3] Scaling Laws for Reward Model Overoptimization in Direct Alignment Algorithms. NeurIPS24.
    [4] Super(ficial)-alignment: Strong Models May Deceive Weak Models in Weak-to-Strong Generalization. ICLR25.
    [5] Alice: Proactive Learning with Teacher’s Demonstrations for Weak-to-Strong Generalization.
    [6] Disentangling Length from Quality in Direct Preference Optimization. ACL24.
    [7] Weak to Strong Generalization for Large Language Models with Multi-capabilities. ICLR25.
    [8] Smaller, Weaker, Yet Better: Training LLM Reasoners via Compute-Optimal Sampling. ICLR25.
    [9] Self-Play Fine-Tuning Converts Weak Language Models to Strong Language Models. ICML24.
    [10] Scaling laws for reward model overoptimization. ICML23.
    [11] Weak-to-strong Reasoning. ACL24.
    [12] How to Mitigate Overfitting in Weak-to-strong Generalization?. ACL25.
    [13] On a Connection Between Imitation Learning and RLHF. ICLR25.

---

### Note · Authors · 2025-08-15

We sincerely thank all reviewers for their constructive feedback. Our paper theoretically reframes the W2S as a **KL-constrained reward-maximization problem** (as appreciated by Rev. EoAi, EDbh, 5hvm, ayzt) and proposes a **novel approach to address the overfitting problem to the weak teacher's mistakes** for **open-ended generation tasks** (as appreciated by Rev. EoAi, ayzt, EDbh), followed by extensive empirical evaluation (as appreciated by Rev. 5hvm, ayzt). We have discussed and addressed:

**(1) Statistical Significance:** Rev. 5hvm and EoAi raised a concern about statistical significance tests. We explained that we **already followed existing protocols of prior works** (e.g., [1-13] in the global response), while providing **statistical significance tests for all experiments in the paper** (with reproducible details) -- a significant effort with large utilization of computational resources. This addressed the concern.

**(2) Additional Evaluation Task:** Rev. EoAi questioned the lack of task diversity. We mentioned that **we evaluated on benchmarks used the related W2S works** [1,2,3,4]. Nevertheless, we still performed evaluations on the suggested **mathematical reasoning task**. EVE again achieved better performance than the baselines -- consistent with the paper's evaluation. This addressed the concern.

**(3) Scaling of large student models:** Reviewer EDbh raised a question about evaluation on students beyond 3B models. We clarified that **our paper already evaluated on 7B student models**, demonstrating that EVE can scale well to larger students. This addressed the concern.

**(4) Theoretical connection to prior W2S works:** Rev 5hvm mentioned the lack theoretical connection to a prior W2S theory work [1]. We explained [1] is orthogonal to ours in contributions, scopes, and objectives, and showed multiple related W2S works ([2-9]) didn't also provide any connection to the theoretical error bound in [1]. We finally detailed why it's extremely challenging to extend [1] in our setting.

**(5) Technical details:** We explained EVE's hyperparameter robustness (we set the same values for all experiments for consistency) and its minimal computational overhead (compared to the baselines)

We believe that all reviewers' concerns have been addressed in our discussions, and we will revise the paper accordingly in its final version.

[1] Lang, H., Sontag, D., and Vijayaraghavan, A., "Theoretical Analysis of Weak-to-Strong Generalization," NeurIPS 2024.

---

### Decision · Program_Chairs · 2025-09-17

**Decision:**

Reject

**Comment:**

The paper studies weak-to-strong generalization, reveals one reason for overfitting to a weak teacher's mistake, and proposes a new method called Weak Teacher Evaluation of Strong Student Demonstrations to regularize the strong student toward its reference policy. After the rebuttal, reviewers are still concerned about the limited evaluation, insufficient baseline comparison, the lack of a precise mathematical formalization of the weak teacher, etc. More efforts are required before the manuscript is ready to publish.